# Modeling and Simulation of Shape Control Based on Digital Twin Technology in Hot Strip Rolling

**DOI:** 10.3390/s24020614

**Published:** 2024-01-18

**Authors:** Youzhao Sun, Jingdong Li, Yamin Sun, Lebao Song, Quan Yang, Xiaochen Wang

**Affiliations:** 1National Engineering Research Center of Flat Rolling Equipment, University of Science and Technology Beijing, Beijing 100083, China; lijingdong_ustb@163.com (J.L.); 18801072578@163.com (Y.S.); s212008522@163.com (L.S.); yangquan@nercar.ustb.edu.cn (Q.Y.); wangxiaochen@ustb.edu.cn (X.W.); 2Institute of Engineering Technology, University of Science and Technology Beijing, Beijing 100083, China

**Keywords:** digital twin, shape model, 3D virtualization, hot strip rolling

## Abstract

Focusing on the problem of strip shape quality control in the finishing process of hot rolling, a shape model based on metal flow and stress release with the application of varying contact rolling parameters is introduced. Combined with digital twin technology, the digital twin framework of the shape model is proposed, which realizes the deep integration between physical time–space and virtual time–space. With the utilization of the historical data, the parameters are optimized iteratively to complete the digital twin of the shape model. According to the schedule, the raw material information is taken as the input to obtain the simulation of the strip shape, which shows a variety of export shape conditions. The prediction absolute error of the crown and flatness are less than 5 μm and 5 I-unit, respectively. The results prove that the proposed shape simulation model with strong prediction performance can be effectively applied to hot rolling production. In addition, the proposed model provides operators with a reference for the parameter settings for actual production and promotes the intelligent application of a shape control strategy.

## 1. Introduction

With the development of processing and manufacturing technology, the iron and steel manufacturing industry has evolved an industrial system integrating resource transformation, energy utilization, quality control and environmental protection. Therefore, it is of great significance for the management and control of steel production from the manufacturing process. Quality control of the hot rolling process mainly depends on strip thickness and shape accuracy. With the application of hydraulic automatic gauge control (HAGC), thickness accuracy has been significantly enhanced. Nevertheless, due to rising demands for product quality, the problem of shape accuracy has become prominent, especially with regard to the strip crown and flatness control in the process of hot rolling.

The essence of shape control involves adjusting the shape of the exit strip’s cross-section and flatness to meet the accuracy requirements by changing the shape of the load roll gap of the rolling mill. Roll profile optimization technology was the earliest adopted flatness control method. Various types of mills have been developed based on mathematical models. For example, a high crown (HC) mill with a work roll bending system was invented to enable control over strip flatness [1]. Subsequently, a universal crown (UC) mill was proposed [2]. In addition, Bald et al. [3] invented the continuous variable crown (CVC) technique, enabling the strip crown to be continuously adjusted within a roll gap. Another type of mill is the pair cross (PC) mill, where the roll axes intersect at an angle to change the shape of the roll gap [4]. The shape model serves as the basis for shape control, and the accuracy of its calculation determines both rolling stability and strip shape. Sun et al. [5] proposed a dynamic programming-based profile distribution model in which the finite element method (FEM) is used for the calculation of key parameters. Moazeni et al. [6] used the FEM to analyze the flow of material and stresses. In addition, Wang et al. [7] adopted a 3D elastic–plastic FEM to analyze the symmetrical flatness actuator efficiencies of a cold rolling mill. In addition, some researchers have established offline or online shape control models based on specific rolling conditions, mainly the characteristics of strip crown and shape actuators, detection instruments and control strategies [8,9,10]. However, the traditional theoretical calculation model is based on specific assumptions and simplified methods that obtain an approximate calculation result with certain errors. The FEM can accurately simulate the actual situation based on the set conditions, but it requires a large amount of computation and takes a long time. With the development of data-driven technology, machine learning models have been introduced into the process of plate profile prediction and control. Ji et al. [11] employed a genetic algorithm to optimize a multi-output support vector regression model for establishing a predictive model of the cross-sectional profile of strip steel. Song et al. [12] employed the particle swarm optimization algorithm to optimize a support vector machine for constructing a predictive model of a hot rolling strip profile. Through comparison with artificial neural networks and regression trees, they validated the feasibility of this approach. Ding et al. [13] employed LightGBM to establish a high-precision and interpretable framework for predicting and diagnosing the crown of hot-rolled strip steel. However, the aforementioned studies focused on the offline prediction of the hot-rolled strip’s shape, making it challenging to achieve synchronous operation with the actual manufacturing process.

In recent years, digital twin (DT) technology has emerged as a prominent trend in technology development, representing physical entities in the virtual world. Constructed DT models not only mirror the attributes of physical objects but also maintain consistency in kinematics and dynamics. In other words, they encompass all the necessary information to characterize the physical manufacturing system and can be continually updated throughout the entire life cycle for continuous and comprehensive monitoring. Data interaction between digital space and physical space makes it easier to integrate with other technologies, such as big data, Internet of Things (IOT), cyber–physical systems (CPS) [14] and artificial intelligence (AI) [15], jointly promoting the development of intelligent manufacturing [16]. In addition, DT technology can also realize the real-time visualization of physical objects to make the manufacturing process more transparent [17]. Compared with physical manufacturing systems, an advantage of DTs is to reduce costs. Especially in the complex product design stage, DTs can be used for design optimization and virtual experiments. For example, in the assembly process of industry, DTs exist in almost all products’ manufacturing processes, and their purpose is to integrate separated parts. At present, DTs have been used to optimize the assembly path [18,19], required force [20], component tracking [21] and task allocation [22] based on human–computer collaboration, minimizing the deformation of parts and improving production efficiency [23,24]. In the field of additive manufacturing, Debroy et al. developed DTs based on numerical simulation to study temperature, metal flow, solidification microstructures and sediment geometric evolution [25]. Large-capacity engineering data sets have been built so as to change the data-driven methods used in traditional experiments, which helps to reduce cost and time. The data used for building DTs are sourced from the information collected by multiple sensors in physical manufacturing systems, so DT technology also has the functions of process monitoring and remote diagnosis. DTs run in sync with the actual manufacturing process, allowing for the easy accumulation and storage of vast amounts of data in the cloud, which can assist big data analysis and pattern recognition based on deep learning [26]. The transition from traditional production management modes to DT-based production monitoring systems is underway. Tong et al. [27] developed a DT application and service utilizing real-time production data. Real-time monitoring becomes essential as the time-varying dynamic conditions caused by wear and aging of parts impact machine accuracy. Erkorkmaz et al. [28] proposed a DT system to monitor the force, torque and motion of multi-axis machine tools, enabling the identification of system dynamics and real-time updates of control strategies to minimize machining errors. Moreover, the collected information can be processed and analyzed to offer machine health assessments [29] and formulate predictive maintenance plans [30].

DTs are one of the core technologies of the industry 4.0 era. They can not only digitize physical entities and provide a virtual object in the physical world, but they also allow this object to possess the same kinematic and dynamic rules. When the virtual model is constantly updated to the real state, the parameters of the DT mechanism model are constantly, iteratively optimized, as well, and finally achieve consistency between the virtual world and physical world. This feature of DTs provides a solution to the transformation of the manufacturing processes of industry like steel plants and to the optimization problems of systematic shape models. Therefore, this paper introduces a systematic shape model to begin with and proposes the system framework of a shape control model based on a DT. With the support of historical data, DT technology is utilized to adjust the shape model parameters to make the simulation results of the product consistent with the actual production data.

The remaining sections are structured as follows: Section 2 describes a new shape model and the establishment of the DT system architecture. Section 3 shows the implementation of the DT, including the establishment and complement of the shape model DT and virtual simulation of the strip crown and flatness under the finish rolling. Section 4 presents the conclusion of this paper.

## 2. System Architecture

We introduce a DT framework that integrates physical and virtual time–space with a data center, facilitating real-time data processing, feature extraction and information fusion to optimize production control. Additionally, we developed variable contact rolling technology and plate profile setting models, establishing the groundwork for modeling and simulating plate profile control.

### 2.1. Structure of Digital Twin

In the hot continuous rolling process, the architecture of the DT is composed of physical time–space, virtual time–space and a data center, as shown in Figure 1. The physical time–space includes the traditional hot rolling strip production control level. Various control and executive elements are used in the basic automation level to directly sense and collect data. In the process control level, the setting of rolling process parameters is completed. According to the production plan, the operator can set the load distribution, rolling temperature and product flatness, etc. In the DT architecture, both the setting data and collected data are transferred to the data center. Data mapping, feature extraction and information fusion can be carried out only after multi-source data are processed. All kinds of data calculated in this process are also stored in the data center. Meanwhile, the processed data are sent to the virtual time–space for synchronous virtual production, which is consistent with the physical time–space and can be used for reference by operators through 3D visualization. State synchronization includes geometric models, dynamic models and kinematic models. Geometric models are proportional models based on the equipment and environment of the workshop, which emphasizes realism. The dynamic models are concerned with the changes in material shape and temperature that occur in the production process and provide an intuitive reference. The kinematic models include material position tracking and equipment motion tracking. With the support of the data center, the interaction between the three models can be successfully completed. A logical judgement is made to continuously modify the production line’s status information to match actual production based on real-time signal change detection.

In addition to the realization of virtual production synchronized with the physical time–space, on the other hand, production simulation becomes possible with the help of various historical production data and state models in the data center. Operators can start the production simulation by providing target production instructions, and the product state changes in the simulation can be intuitively observed from the 3D virtual production. At the same time, all kinds of process data generated are recorded and presented to the operators in a reasonable way to provide a reference for the arrangement of the production plan. The parameter settings of the production simulation are optimized by iteration, and the most suitable parameter-setting scheme of the production line is given. In addition, operators can reasonably adjust the parameter settings of the real production line in physical time–space.

### 2.2. Varying Contact Rolling

Hot rolling is the final process before the production of hot-rolled strips, and this process aims to improve the surface quality of the strip, optimize the shape of the plate profile, improve the strip’s mechanical properties and eliminate defects such as strip waves. Shape quality is a significant quality indicator for hot rolling products and a meaningful research object for hot rolling production. The shape model is influenced by many factors, which can be divided into two types. One is factors related to the condition of the roll, such as roll shifting, roll bending, roll wearing and rolling force fluctuation. On the other hand, the shape model can be influenced by the factors of the strip itself, such as temperature, entry thickness and width, target thickness and width and elastic modulus.

In order to eliminate or reduce the impact of rolling force fluctuations and changes in roll shape on shape control and operation and improve the shape control performance of the rolling mill, this work uses varying-contact-length backup roll (VCR) technology in the hot strip rolling mill. The VCR adopts a specially designed roll shape curve, based on the production characteristics of hot continuous rolling, that can effectively enhance the flatness control performance of the rolling mill and improve the strip quality [31]. In addition, the VCR also possesses excellent performance characteristics, such as improving the maximum contact pressure between rolls to avoid the spalling of the backup roll.

### 2.3. Establishment of Shape Model

The difficulty lies in the coupling of different factors, as the model parameters will change under complex production conditions and different production schedules. Therefore, determining the methods for updating the parameters adaptively becomes the key problem in the application of the shape model. As shown in the DT architecture of Figure 1, the shape model is an important component of its dynamic model. Unlike traditional shape models (assuming no material flow in the strip width direction and ignoring the recrystallization of the rolled piece between the stands), the adopted VCR model considers the lateral flow of the material that occurs during the deformation of the roll gap, and its calculation process is shown in Figure 2.

Undoubtedly, a good shape is obtained by adjusting the control parameters of each stand, and the core process is to adopt a certain shape preset strategy. In contrast to the traditional plate shape preset strategy, the empirical coefficient is used to allocate the proportional crown of the upstream stands to achieve the target proportional crown, and the downstream stands are strictly controlled according to the principle of equal proportional crown allocation [32]. The principle of the proportional crown is controlled. The shape preset model used in this system introduces the residual stress influence factor ζ (Equation (1)) to characterize the control ability of each stand. In addition, a shape preset strategy based on the residual stress factor is proposed to allocate the proportional crown of each stand (Equation (2)), so as to calculate the required roll gap crown of each stand and complete the corresponding roll bending force and realize the setting calculation of the roll shift value [33].

Because of the residual stress caused by the change in the proportional crown, it will attenuate due to the lateral flow of metal and the stress relaxation effect between the stands during the primary and secondary deformation stages [34]. In the case of considering the mechanical characteristics of the strip, the influence factor of the residual stress ζ is defined as follows:(1)$$\zeta =\frac{\beta_{sr,i}}{\beta_{f}E}$$
where *β_sr,i_* is the stress relaxation coefficient of the strip running at the stand *i*, *β_f_* is the metal lateral flow coefficient and *E* is the elastic modulus of the strip material.

On the basis of the residual stress influence factor, when the proportional crown of the finish rolling entrance strip and the proportional crown of the finishing strip exit are given, the entrance proportional crown of each stand can be determined by the following formula:(2)$$C_{Peff-H,i}=C_{Peff-h,i}+\frac{\zeta_{i}}{\sum_{1}^{i}\zeta_{i}}(C_{Peff-H,i}+C_{Peff-h,i})\;(i=1\sim 7)$$
where *C_Peff-H,i_* is the effective proportional crown of the entry strip for finish rolling and *C_Peff-H,i_* is the effective proportional crown of the finish rolling exit.

In the calculation stage of the shape control parameter setting, the roll bending force and shifting value for each stand are sequentially computed, starting with the last stand. The main process is as follows: First, calculate the required target uniform roll gap crown based on the effective exit proportional crown and the effective entrance proportional crown, calculated according to the proportional crown allocation; then, according to the target uniform roll gap crown, combine the initial values, including bending force, roll shifting value, rolling force and other parameters, to calculate the required target integrated roll system crown; finally, calculate the corresponding roll shifting value according to the target integrated roll system crown. When the position of the roll shifting is determined, the crown of the integrated roll system is recalculated according to the setting roll shift position, and then the roll bending force is calculated. Finally, judge whether the proportional crown of the uniform load roll gap obtained by the mechanical adjustment structure meets the target requirement. If the requirement cannot be met, calculate the residual stress at the exit of the roll gap. When the residual stress does not meet the critical buckling limit, reset the target crown, calculate the effective proportional crown of the exit and entrance at the final stand and restart the circulation process.

## 3. Implementation

We employed a 2250 mm hot rolling production line from a steel plant as a case study, establishing a digital twin production line. Through simulation experiments on its products, we confirmed the efficacy of the proposed methodology outlined in this paper.

### 3.1. Industrial Context

Based on a 2250 mm hot strip mill project, this paper establishes a shape model based on a DT. The layout of the 2250 mm hot strip mill is shown in Figure 3, in which the main process equipment, including three furnaces, one slab sizing press, two four-high roughing mills and one hot strip box, is preserved, and one edge heater, one crop shear, seven four-high finishing mills, one set of laminar strip cooling devices, three underground coilers, one coil transportation line with a sampling inspection device, one leveling and winding line and auxiliary facilities matching the production line, such as an electrical automation system, water treatment plant, hydraulic lubrication, etc., are added. The products from this hot rolling line span a thickness range of 1.2 mm to 25.4 mm for plates, with strip widths varying from 800 mm to 2130 mm. The primary steel types include carbon structural steel, medium carbon alloy steel, steel for automotive structures, bridge steel and pipeline steel, among others. The slab casting temperature falls within the range of 1150–1300 °C. These data are of considerable significance for refining the simulation calculation of shape model parameters.

The strip crown and flatness are two important aspects used to describe the strip shape, which is one of the most important geometric dimensions besides the mechanical properties of the strip itself.

As shown in Figure 4a, the strip crown (*C_h_*) refers to the difference between the thickness *h_c_* at the center of the strip and the average thickness at the edge marker points (*h_L_* and *h_R_*), defined as follows:(3)$$C_{h}=h_{c}-\frac{h_{L}+h_{R}}{2}$$
where *h_c_* is the thickness of the center of the strip, mm; and *h_L_* and *h_R_* are the thicknesses at the marked points on the operation side and driven side of the strip.

As shown in Figure 4b, flatness is an indicator that describes the longitudinal deformation of the strip, mainly used to indicate whether the strip has flatness defects. This work uses the relative length ∆*l*/*l* to represent the strip flatness, and the expression is as follows:(4)$$Flt=\frac{\Delta l}{l}\times 10^{5}$$
where *h_c_* is the strip flatness, I; *l* is the length after rolling of the reference point, mm; and Δ*l* is the difference in length between other points and the reference point after rolling, mm.

The finishing mill adopts a CVC mill, which performs the functions of work roll bending and roll shifting. The main parameters of the finishing mill are shown in Table 1. These parameters are an important basis for the establishment of the DT.

### 3.2. Construction of Digital Twin

To give full play to the shape control ability of each rolling mill and effectively control the strip crown, the DT of the shape model proposed in this paper was applied to a 2250 mm hot continuous rolling line to simulate the strip shape. The first step in the construction of the DT requires the modeling of physical objects, which is shown in Figure 5. The virtual scene contains 3D models of production line equipment and materials, and the construction of each 3D model includes building a 3D white model and setting the material type. According to the 2D drawings of the equipment in the hot rolling line, the basic model was established in 3DS max by using the basic body, and then the 3D white model was created by using Boolean, mesh, polyhedron and other editing tools. To ensure the simulation had more fidelity, the appearance material of the 3D model reflecting the characteristics of the production site was drawn with reference to photos or videos. On the other hand, according to the production process, process principle and equipment function, the dynamic tracking of the materials was established to ensure the simulation better reflected real-time conditions.

To make the proposed shape model fit for the actual producing requirements, according to the device instructions, the linear and horizontal motion of the device was set in the modeling software, which is the main form of motion in the hot rolling process. Moreover, the movement state of the strip was mainly set according to the speed signals from the collected data. As the model is an equal-scale replica of the real object, the virtual model is more intuitive. The model of the finishing mill is shown in Figure 6.

### 3.3. Simulation and Results

Firstly, a large amount of detailed historical production data was obtained from the steel plant, which is the key to driving the DT. In order to make the shape model meet the actual production requirements, the historical production data were taken as the input of the DT. To obtain the corresponding strip shape results, the DT can update iteratively until the results meet the production requirements. After repeated simulations of the real production data, the DT of the shape model was completed, and it was then added to the shape simulation prediction. The following section shows the DT of the shape model simulating the rolling production of the finishing mills.

Figure 7a shows the effect of simulating roll bending based on VCR technology. Figure 7b–d show the shifting roll based on VCR technology and different methods of shape control. In addition, the effects of hot strip flatness are shown in Figure 8.

In addition, combined with the production schedule, the rolling effects of the hot strip were simulated. This experiment involved 100 steel coils, with the relevant parameters outlined in Table 2. The comparison between the simulation test results from the proposed DT model and the actual newly rolled results is depicted in Figure 9. In Figure 9a, the model-predicted results for the average crown of strip C40 are presented, while Figure 9b illustrates the predicted outcomes for the head crown of strip C40. Additionally, Figure 9c displays the predicted results for the tail crown of strip C40, and Figure 9d showcases the model’s predictions for strip flatness.

As shown in Figure 9, the strip shape prediction results, including crown and flatness, are concentrated near the 45° diagonal, and the absolute error between the predicted strip crown values and the actual measured values is less than 5 μm. The absolute error between the predicted flatness results and the actual measured results is within 5 I-units. To more effectively validate the accuracy of the model, we used root mean square error (RMSE) and R-squared (R^2^) for the quantitative evaluation of the model’s precision. It can be observed that the RMSE values for the DT model’s prediction of the average crown, head crown, tail crown and flatness of the strip are 2.18 mm, 1.95 mm, 1.78 mm and 2.37 I-units, respectively. As for R^2^, the DT model achieved values of 0.83, 0.93, 0.92 and 0.97 for the average crown, head crown, tail crown and flatness, respectively. The results fully proved that the proposed shape simulation model with strong learning ability and prediction performance can be effectively applied to hot rolling production. In addition, the shape simulation of the DT provides operators with a reference for the parameter settings of actual production and promotes the intelligent application of a shape control strategy.

## 4. Conclusions and Outlook

Aiming to address the problem of shape control in hot strip rolling, this paper researches a digital twin shape simulation model and realizes the integration of physical time–space and virtual time–space. The digital twin architecture of the finish rolling shape model is presented, and the shape simulation of the finish rolling process is realized, which makes up for the lack of production strategy evaluation. The main conclusions are as follows:(1)The shape model introduced considers the metal flow and the stress relaxation between stands; the residual stress influence factor is proposed, and the shape preset strategy is given. Compared with the traditional shape preset strategy, this model can give full play to the control ability of each stand with the application of varying contact rolling.(2)Combined with digital twin technology, the digital twin framework of the shape model is proposed, which realizes the deep integration between physical time–space and virtual time–space. With the support of an actual industrial background, the parameters of the shape model are updated iteratively by utilizing historical production data, and the digital twin of the shape model is completed.(3)Combined with the production schedule, the rolling effects of the hot strip are simulated. The raw material information is used as input, and different shape results are obtained, which can assist the operator when modifying and improving the presetting operation. The prediction absolute errors of crown and flatness are less than 5 μm and 5 I-units, respectively. These results prove that the proposed shape simulation model with strong prediction performance can be effectively applied to hot rolling production. In addition, the proposed model provides operators with a reference for the parameter settings of actual production and promotes the intelligent application of a shape control strategy.

This paper focuses on shape control in the finishing process. The shape model introduces the concept of influence factors to characterize the control ability of finishing mill stands and comprehensively considers the transverse flow of strip metal and the residual stress relaxation effect between stands. The digital twin technology is effectively combined with actual hot rolling production. However, in other hot rolling processes, such as laminar cooling, it affects the shape as well. Therefore, the follow-up work will comprehensively consider the shape change mechanism, and the digital twin technology should be applied to the entire hot strip rolling line to realize the intelligent control of mass flow in the entire hot strip rolling process and further improve the intelligence level of hot rolling.

## Figures and Tables

**Figure 1 sensors-24-00614-f001:**
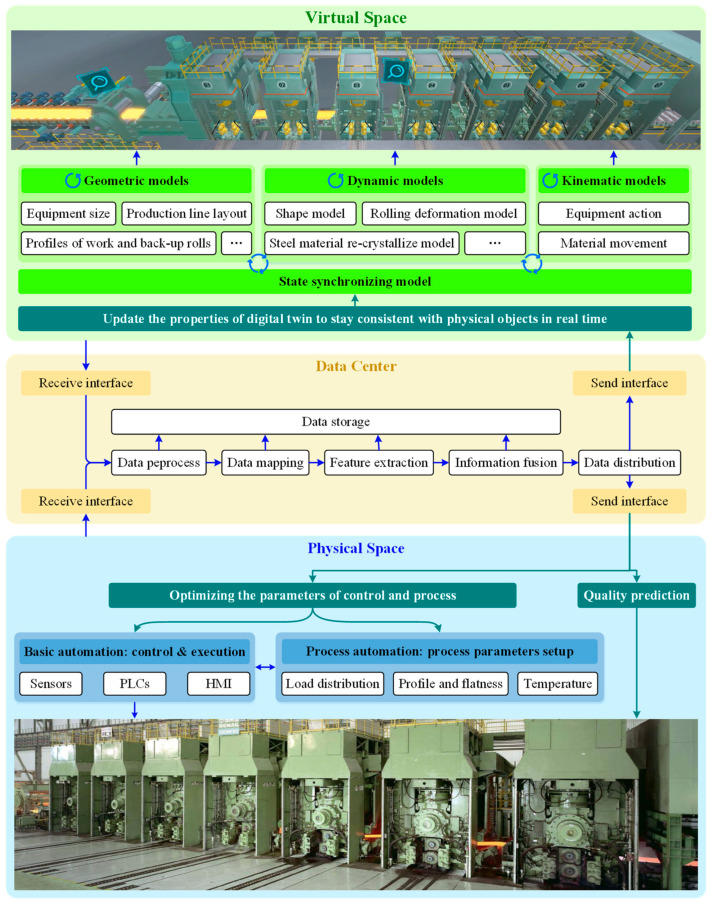
DT architecture of hot strip mills.

**Figure 2 sensors-24-00614-f002:**
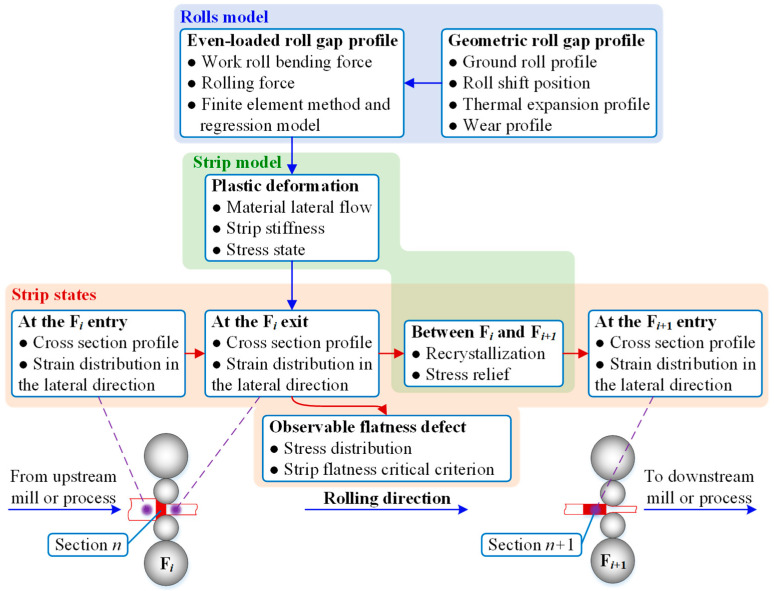
The calculation flow of the shape model.

**Figure 3 sensors-24-00614-f003:**
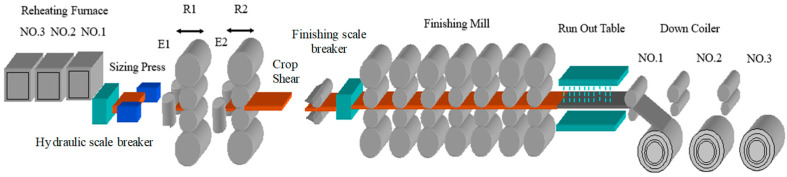
Hot strip rolling product line.

**Figure 4 sensors-24-00614-f004:**
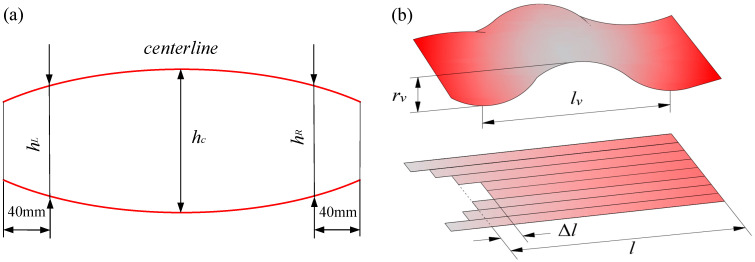
Description of the strip shape: (**a**) crown and (**b**) flatness.

**Figure 5 sensors-24-00614-f005:**
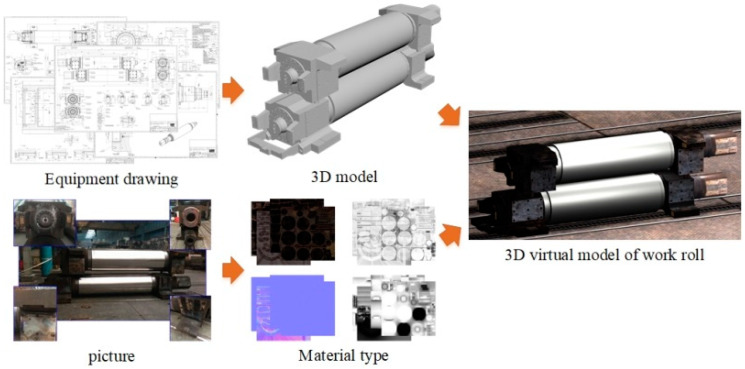
Establishment of 3D equipment model.

**Figure 6 sensors-24-00614-f006:**
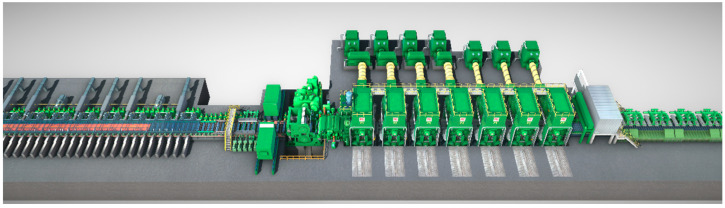
The 3D model of the finishing mill line.

**Figure 7 sensors-24-00614-f007:**
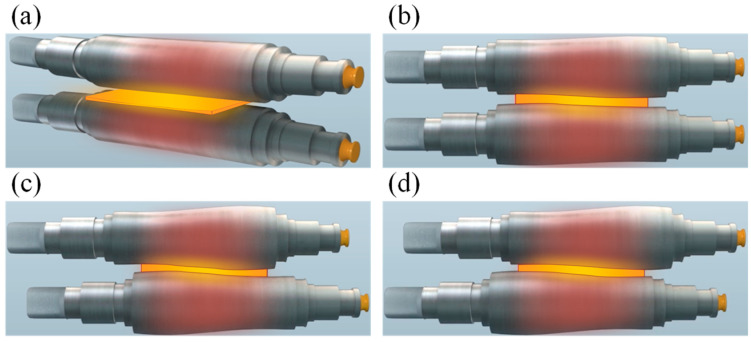
(**a**) Bending roll based on VCR, (**b**) central crown, (**c**) positive crown and (**d**) negative crown.

**Figure 8 sensors-24-00614-f008:**
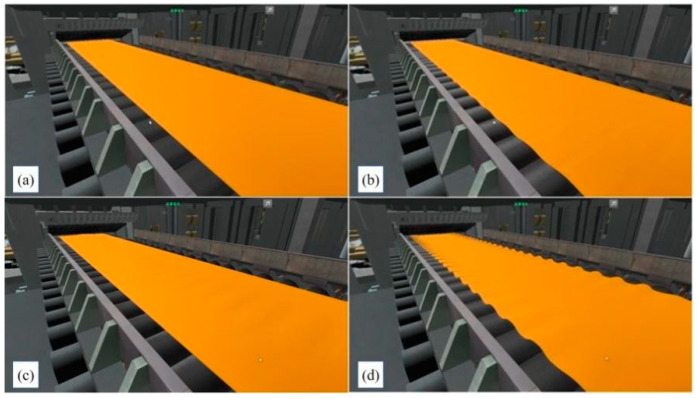
Visualization of flatness defects. (**a**) normal strip, (**b**) single edge wave, (**c**) central wave, (**d**) bilateral wave.

**Figure 9 sensors-24-00614-f009:**
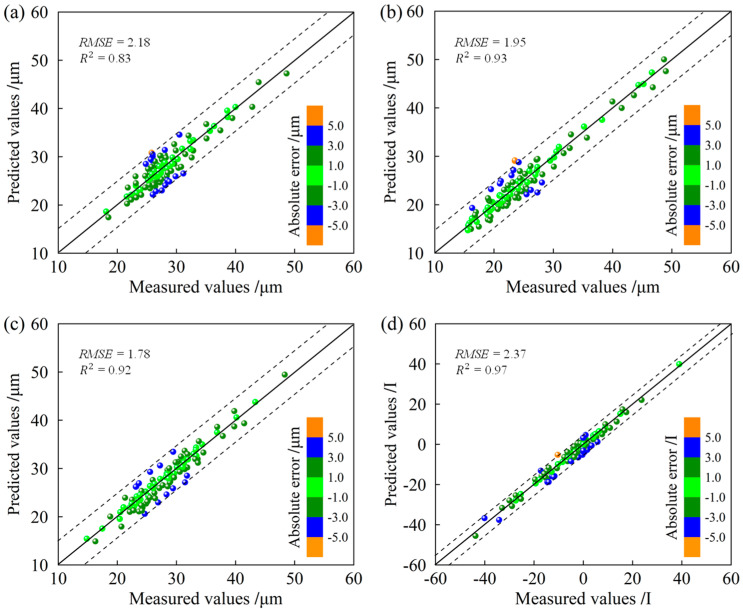
Results of the DT model: (**a**) average crown, (**b**) head crown, (**c**) tail crown and (**d**) flatness.

**Table 1 sensors-24-00614-t001:** Dimensions of finishing mill.

Parameter	Stand	Value	Unit
Backup roll length	F1~F7	2250	mm
Work roll length	F1~F7	2250	mm
Work roll shifting	F1~F7	±150	mm
Backup roll diameter	F1~F7	1600/1440	mm
Work roll diameter	F1~F4	850/765	mm
	F5~F7	760/685	mm
Work roll bending force	F1~F7	1500	kN
Max rolling force	F1~F4	50,000	kN
	F5~F7	40,000	kN

**Table 2 sensors-24-00614-t002:** Description of experimental strip parameters.

Steel Grade	Parameters	Range	Unit
SPHC-W1	Slab thickness	230	mm
	Finish rolling thickness	2.3–3.5	mm
	Finish rolling entry temperature	980–1040	°C
	Finish rolling exit temperature	850–906	°C

## Data Availability

Data are contained within the article.

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
