# Peer review of "Modeling and Simulation of Shape Control Based on Digital Twin Technology in Hot Strip Rolling"

_sensors, 2024, doi:10.3390/s24020614_

Round 1

Reviewer 1 Report

Comments and Suggestions for Authors

1.       The introduction needs to go much further, the authors only introduces other works very simply, the reviewer can not see the deep relationship between references and this work.

2.       Writing needs improvement. Each section should open with a paragraph or two explaining what to expect before going to subsections.

3.       In section 3.3, the simulation model set up details should be presented.

4.       In Fig.10, more detail parameters such as R and R2 should be given.

5.       The reviewer wonders how does the real residual stress be calculated and measured? How about the error level?

The depth of technical discussions needs to go much further.

Comments on the Quality of English Language

Writing needs improvement. Each section should open with a paragraph or two explaining what to expect before going to subsection

Reviewer 2 Report

Comments and Suggestions for Authors

1. Please, add more information concerning 2250 mm hot strip mill. For instance, location, plant, etc. What are the steel grades used for rolling in this mill? What is the temperature range?

2. It seems reasonable to explain some abbreviations of Fig. 3 in the figure caption (for reader convenience): FSB, HSB, etc.

3. Please, specify the regime(s) for which Fig. 10 results were obtained. What were the initial and final thickness, what was the temperature of the coil prior to rolling?

Reviewer 3 Report

Comments and Suggestions for Authors

This manuscript is devoted to the solution of an urgent technical problem, namely the creation of a digital twin of the hot strip rolling process. This solution will increase the efficiency of production by predicting the occurrence of defects and improving the accuracy of products. The article describes all stages of the model development in sufficient detail. At the same time, the manuscript contains some inaccuracies that should be eliminated before publication.

I believe that this manuscript can be published after minor revision.

The main comments and recommendations are as follows:

(1) What are the sensors and transducers used in the physical system to monitor the hot rolling parameters.

(2) Please combine the figures 7 and 8.

(3) page 7, lines 229-230. Repetition of words.

(4) Please check the description to formula (4).

(5) It is not clear from the description of the manuscript whether the results predicted by the model were compared with new experimental rolling results or only with historical data.

(6) How, and for what purpose, is the recrystallization model taken into account in the proposed digital twin model?

Comments on the Quality of English Language

The manuscript contains minor typos and inaccuracies.

The English is sufficiently correct and written in a technically competent manner.

Round 2

Reviewer 2 Report

Comments and Suggestions for Authors

In my view, necessary corrections were made.